# Focal Active Colitis: What Are Its Clinical Implications? A Narrative Review

**DOI:** 10.3390/biomedicines11102631

**Published:** 2023-09-25

**Authors:** Emanuele Sinagra, Francesco Vito Mandarino, Marcello Maida, Daniela Cabibi, Francesca Rossi, Dario Raimondo, Guido Manfredi

**Affiliations:** 1Gastroenterology and Endoscopy Unit, Fondazione Istituto G. Giglio, Contrada Pietra Pollastra Pisciotto, 90015 Cefalù, Italy; fraross76@hotmail.com (F.R.); dario.raimondo@hsrgiglio.it (D.R.); 2Division of Gastroenterology and Gastrointestinal Endoscopy, Department of Experimental Oncology, IRCCS San Raffaele Scientific Institute, Vita-Salute San Raffaele University, 20132 Milan, Italy; mandarino.francesco@hsr.it; 3Gastroenterology and Endoscopy Unit, S. Elia-Raimondi Hospital, 93100 Caltanissetta, Italy; marcello.maida@icloud.com; 4Pathology Unit, Fondazione Istituto G. Giglio, Contrada Pietra Pollastra Pisciotto, 90015 Cefalù, Italy; danielacabibi@virgilio.it; 5Gastroenterology and Digestive Endoscopy Department, ASST-Crema Maggiore Hospital, 26013 Crema, Italy; guidomanfredi@virgilio.it

**Keywords:** focal active colitis, Crohn’s disease, ulcerative colitis, self-limiting colitis, indeterminate colitis

## Abstract

Focal active colitis (FAC) is described as a histolopathological term indicating the isolated finding of focal neutrophil infiltration in the colonic crypts. Currently, there exist numerous debates regarding the clinical significance of diagnosing FAC, which may or may not have clinical relevance as it is frequently detected in colorectal biopsies without any other microscopic abnormalities. The objective of this narrative review is to provide an overview of the available evidence concerning the clinical implications of FAC, both in the adult population (among five studies available in the scientific literature) and in the pediatric context (based on two available studies).

## 1. Introduction

Focal active colitis (FAC) is a histologic term which denotes the presence of focal neutrophil infiltration within the colonic crypts [1]. To date, there has been significant debate regarding the clinical implications of FAC diagnosis, as it is frequently identified in colorectal biopsies without other notable microscopic abnormalities [2,3].

This entity was initially documented by Greenson and colleagues, who studied a group of 49 patients and found infections and drugs to be the most common causes. They also noted no clear association with the pathological findings and inflammatory bowel diseases (IBD) [2]. The histological anomalies they observed varied, ranging from solitary crypt abscesses or isolated cryptitis to multiple distinct foci of cryptitis or crypt abscesses within colorectal biopsies [2,3]. Notably, no substantial signs of chronic inflammation were present in FAC samples [2].

Traditionally, focal cryptitis has been used interchangeably to describe this phenomenon, primarily because it was believed to be a moderately reliable indicator for diagnosing Crohn’s disease (CD), particularly in distinguishing it from ulcerative colitis (UC) [2,3].

Subsequent studies have indicated a low incidence of CD among adults diagnosed histologically with FAC [1,2,4]. Furthermore, despite several FAC cases in adults being linked to infectious origins, specific pathogens were not identified in most cases [3]. Notably, FAC was also observed in patients with irritable bowel syndrome (IBS). This narrative review aims to summarize the available evidence regarding the clinical implications of FAC, both in adult and pediatric populations.

## 2. Focal Active Colitis in the Adult Setting

According to the initial description by Greenson and colleagues, FAC was considered a potential feature of CD but could also be found in cases of colonic ischemia, infections, partially treated UC, and even as an isolated finding in patients undergoing endoscopy to rule out neoplasia [2]. In their study, they retrospectively analyzed clinical, endoscopic, and pathological data from 49 FAC patients who had no other identifiable findings on colorectal biopsy and no history of chronic IBD. Histological findings were correlated with clinical diagnoses, with follow-up information available for 42 of these patients. Interestingly, none of them developed IBD; instead, 19 patients experienced acute self-limited colitis-like diarrhea, 11 had incidental FAC (detected during endoscopy in asymptomatic patients), 6 were diagnosed with IBS, 4 had antibiotic-associated colitis, and 2 had ischemic colitis. Importantly, histological features did not predict the final diagnoses conclusively [2]. The authors concluded that most FAC cases, especially those with acute self-limited colitis and antibiotic-associated colitis, were likely infectious in nature. The incidental detection of FAC in patients undergoing neoplasia screening was not clinically significant [2].

A separate study by Volk and colleagues assessed 31 cases of FAC at the Cleveland Clinic Foundation between 1982 and 1992 to establish the clinical significance of this histological finding [4]. They also evaluated the extent of neutrophil-mediated crypt epithelial injury, surface epithelial injury, and lamina propria cellularity and cell type. Each biopsy specimen was scored as 1+ (<10%), 2+ (10–25%), or 3+ (26–50%) for the mentioned features, with a mean clinical follow-up period of 26 months [4]. Clinical diagnoses included infectious-type colitis (15 cases, 48%), incidental FAC (9 cases, 29%) in asymptomatic individuals screened for colorectal cancer (CRC), ischemic colitis (3 cases, 10%), and CD (4 cases, 13%) [4]. Histologically, all cases displayed some degree of cryptitis, with 24 (77%) out of the 31 cases showing neutrophil-mediated surface epithelial injury [4]. Additionally, 13 cases (42%) had an expansion of the lamina propria with neutrophils, 12 (39%) with eosinophils, and 11 (35%) with plasma cells. None of these histological features strongly correlated with specific clinical diagnostic categories [4]. The authors concluded that FAC most clinically correlated with infectious-type colitis and occasionally served as an early indicator of CD. Furthermore, they found that histological features were not reliable in predicting specific clinical diagnoses associated with FAC [4].

Shetty and colleagues conducted the largest and only prospective series on FAC, examining 90 patients to assess clinicopathological correlations. This study included comprehensive clinical follow-up and questionnaires, and it implicated drugs, especially nonsteroidal anti-inflammatory drugs (NSAIDs), in 24% of cases. Infections were responsible for 19% of cases. Among 14 patients (15.6%), mainly women, a diagnosis of chronic IBD, predominantly CD, was eventually established. Additionally, a specific subtype of FAC known as basal FAC was significantly associated with drug use. However, beyond these findings, the study failed to identify other histopathological parameters of FAC, such as quantity, location, or distribution, that reliably correlated with clinical outcomes or aided in selecting patients more likely to develop subsequent evidence of chronic IBD.

Our research team conducted a retrospective study using prospectively collected data to assess the clinical implications of FAC in 30 consecutive patients (11 males, 19 females; age range 24–80 years, median age 56 years) representing 0.5% of 5600 individuals who underwent colonoscopy between January 2012 and December 2016 [5]. The follow-up period ranged from 6 to 60 months, with a median of 24 months [5]. Endoscopically, 19 patients (63%) exhibited mild and nonspecific changes, such as mild mucosal erythema, while 11 (37%) had normal findings. Eight patients were diagnosed with IBS, nine cases were attributed to drug effects, five presented FAC as an incidental finding, one was diagnosed with infectious colitis, and seven were diagnosed with IBD (four with Crohn’s disease) [5]. Our study emphasized that FAC was a more significant predictor of IBD than previously reported in the literature, although larger prospective studies designed to investigate this relationship are necessary for a clearer understanding of its clinical significance [5].

Hosack and colleagues conducted a retrospective study to evaluate whether FAC is also a reliable predictor of developing IBD in adults [6]. They included 43 patients (11 males, 32 females, mean age 53 years ± 18) with a mean follow-up period of 36 months [6]. Some 14 patients (33%) were diagnosed with infective colitis, and 5 (12%) were subsequently diagnosed with IBD, including 4 (80%) with UC and 1 (20%) with undetermined IBD. Additionally, of the 34 patients (79%) with neither elevated fecal calprotectin (FC) levels nor suspicious endoscopic findings, only 1 (3%) patient later developed IBD. This was statistically significant according to Fisher’s exact test (*p* = 0.0046), and the phi coefficient of 0.53 demonstrated that patients with neither raised FC levels nor suspicious endoscopic findings were statistically unlikely to develop IBD [6]. Interestingly, the authors concluded that having normal FC levels and endoscopic findings reduces the risk of future development of IBD in adults, while higher FC levels and endoscopic features suggestive of IBD with histological FAC may predict progression to IBD [6].

Taken together, these pieces of evidence, despite the methodological limitations of most of these studies, underscore the importance of raising awareness about FAC and its potential evolution into a more aggressive disease like IBD. Larger prospective studies aimed at investigating the relationship between FAC and IBD are warranted to gain a clearer understanding of its clinical significance. Table 1 displays all the diagnoses associated with the occurrence of FAC in the available studies in the adult setting.

## 3. Focal Active Colitis in the Pediatric Setting

In contrast to adults, there are limited available data on the clinical course of FAC in pediatric patients. Only two studies have provided information in this setting. In a study conducted by Xin and colleagues, the authors reviewed 31 cases of FAC diagnosed at pediatric ages with no prior history of IBD between 1989 and 2000. In this cohort, eight patients (27.6%) later developed Crohn’s disease. Nine patients (31%) exhibited symptoms consistent with acute infectious-type colitis, with one case attributed to C. difficile infection. Eight patients (27.6%) had FAC that did not correlate with their symptoms or ultimate clinical diagnosis; these cases were labeled idiopathic FAC. Additionally, two patients were diagnosed with allergic colitis, one with UC, and one with Hirschsprung’s disease. Pediatric patients with FAC displayed a significantly higher incidence of CD compared to adults with the same condition. Therefore, the authors stressed the importance of documenting the presence of FAC in pediatric patients [7].

Osmond et al. and colleagues conducted a retrospective study involving 68 pediatric patients with FAC, which revealed that 16 patients (24%) received a final diagnosis of IBD. When cases with inflammation limited to the terminal ileum (TI) were excluded, 6 out of 54 patients had a final diagnosis of IBD (11%). Furthermore, a final diagnosis of IBD was significantly associated with crypt abscesses and elevated serum inflammatory markers. IBD was also significantly associated with TI inflammation. However, an amount or pattern of inflammation that could be used to predict IBD was not determined. This study demonstrated a 24% rate of IBD in pediatric patients with FAC; however, when patients with associated TI inflammation were excluded, the rate was 11%, similar to reported rates in adults. Therefore, the authors concluded that FAC in pediatric patients without terminal ileal inflammation does not appear to warrant more aggressive follow-up [8].

Based on this evidence, it is suggested that for pediatric patients a comment in the pathology report similar to that used in adult cases of FAC should be added. Specifically, the comment should mention that FAC may lack clinical significance or may be associated with factors such as bowel preparations, acute infections, drugs, and occasionally IBD, unless the presence of crypt abscesses or TI inflammation is observed. In this case, additional follow-up may be suggested [8]. Table 2 presents all the diagnoses associated with the occurrence of FAC in the available studies in the pediatric setting.

## 4. Clinical Implications of Focal Active Colitis

Colonic injury and inflammation may have various underlying causes. The most frequent forms of colitis encountered in clinical practice are those associated with infections, IBD, ischemia, radiation exposure, and medications.

FAC is characterized by isolated neutrophilic cryptitis, with the surrounding mucosa maintaining normal crypt architecture. This inflammatory pattern can sometimes go unnoticed by pathologists because on low-power examination, the mucosa may appear almost normal.

Multiple different conditions can manifest as FAC when examined in pathology reports. The most frequent scenarios include acute self-limited infectious colitis, early stage IBD, IBS, ischemic colitis, Clostridium difficile colitis, medication or chemical-induced injury, and artifacts from bowel preparation [9,10]. The FAC pattern is typically not observed in UC; when it is present, it raises the suspicion of a diagnosis of Crohn’s colitis or infectious colitis and/or acute self-limited colitis. In particular, FAC can be stigmata or precursor lesion of CD or not. As aforementioned, interestingly, Hosack and coworkers concluded that having normal FC levels and endoscopic findings reduces the risk of future development of IBD in adults, while higher FC levels and endoscopic features suggestive of IBD with histological FAC may predict progression to IBD. On the other hand, in the pediatric setting, since IBS and drug-induced colitis are uncommon in children, a higher number of pediatric patients with the inflammatory pattern of FAC in colonic biopsies are likely to have IBD.

However, FAC patterns can be observed in cases of UC that are resolving under medical treatment, and areas of the colon and rectum that were previously inflamed in UC can return to an almost normal histological appearance. Medications associated with FAC include nonsteroidal anti-inflammatory drugs (NSAIDs), mycophenolic acid, and ipilimumab, among others [11,12]. While diarrhea is the most common reason for a workup in many of these patients, FAC can also be incidentally discovered because of bowel preparation with Phospho-soda in asymptomatic patients undergoing screening colonoscopy [6]. Ozdil et al. reported that 6.6% of patients with IBS had FAC, making it reasonable to consider routine biopsies in female patients and those aged above 50 years [13]. Microscopic colitis was more frequently found in the diarrhea-predominant or mixed subgroups of IBS patients and in older female patients. Therefore, in older women with non-constipated IBS, performing a biopsy to screen for microscopic colitis may be considered. FAC remains a diagnostic challenge, and clinicians must consider the patient’s medical history, medication use, and findings from endoscopy to help narrow down the potential underlying causes. Histological examination can provide valuable insights; however, there is no specific finding or feature that can definitively distinguish between incidental and clinically significant FAC, such as a specific minimum neutrophil count.

## 5. Differential Diagnosis

### 5.1. Healthy Colonic Mucosa

Normally, in healthy colonic mucosa, we find evenly distributed straight crypts that extend from the luminal surface to the muscularis mucosae [14]. In this context, goblet cells are less abundant in the right colon compared to the left colon, particularly in the sigmoid colon and rectum, while inflammatory cells predominate in the luminal third of the lamina propria [14].

As of now, there are no established classifications for the histologic patterns found in colonic biopsy samples from patients with acute colitis (AC) [14]. However, two fundamental patterns can be identified based on the underlying disease pathophysiology: inflammatory and ischemic patterns [14]. The former pattern is characterized by the invasion of the mucosa by inflammatory cells, while the latter is defined by damage resulting from hypoperfusion [14]. Worsened perfusion can stem from circulatory impairment, such as low-flow conditions, or from inflammatory/toxic endothelial damage, with or without occlusive fibrin thrombi. In the latter case, it may be associated with variable degrees of inflammatory injury [14].

Therefore, these patterns are not mutually exclusive [14].

FAC, along with AC or acute self-limited colitis, is considered a pure inflammatory colitis. Pseudomembranous colitis (PMC), along with hemorrhagic colitis (HC), is classified as a mixed inflammatory/ischemic colitis. Conversely, acute ischemic colitis (AIC) is regarded as a pure ischemic colitis [14].

### 5.2. Acute Colitis

In cases of AC, preserved crypt architecture and a distribution of inflammatory alterations that can be either diffuse or patchy can be observed [14]. The lamina propria usually appears edematous, with mixed inflammatory cell infiltrates and abundant neutrophils [14]. Inflammatory cells infiltrate the crypt epithelium, leading to luminal aggregates (crypt abscesses) that result in crypt destruction, as well as erosions or ulcerations of the mucosa [14]. Additionally, hemorrhage in the lamina propria may be present. The loss of epithelial cells triggers a regeneration process characterized by increased mitotic activity and the proliferation of immature cells without significant crypt architectural remodeling [14]. In subsequent stages of the condition, neutrophils become less abundant, while plasma cells and eosinophils are recruited to the lamina propria [14].

### 5.3. Early Onset Inflammatory Bowel Disease

It is important to differentiate AC from early onset IBD, particularly when specimens are obtained shortly after the initial onset of symptoms [14]. However, many patients do not undergo colonoscopy during the early stages of the disease [14].

Typically, patients with atypical symptoms or those who do not respond to antibiotic therapy undergo colonoscopy with mucosal biopsy [14]. In such cases, the biopsy specimens often contain plasma cells, eosinophils, and macrophages, along with only a few neutrophils, making differential diagnosis between the diseases challenging [14].

In these situations, this diagnostic challenge can be resolved with the aid of clinical information [14,15]. Even in its early phases, UC is associated with dense, transmucosal inflammation rich in plasma cells within the lamina propria, along with crypt remodeling [14]. In contrast, the plasma cell-rich inflammation of AC usually exhibits a patchy distribution and rarely extends into the deep lamina propria [14].

Conversely, distinguishing between AC and CD poses a more significant challenge. Salmonella and Yersinia infections usually affect the ileocecal area and appendix, mimicking the distribution of CD [14]. Furthermore, as in CD with colonic involvement, AC of any etiology often displays a patchy mucosal distribution [14]. CD with colonic involvement generally produces infrequent crypt architectural changes, and granulomas are typically absent [14]. The features that often favor a diagnosis of AC over CD include the presence of disproportionately more neutrophils in the lamina propria than in crypts, and colonic inflammation unaccompanied by ileal involvement [14,15].

### 5.4. Drug-Related Colitis, with Particular Attention to NSAIDs

NSAIDs can also induce acute ileitis and/or colitis, resembling those seen in CD [14,16,17,18]. Well-defined ulcers in the right colon, surrounded by healthy-looking mucosa, are frequently observed in NSAID-related injury, although any part of the colon may be affected [14]. Granulomas are not typically found in AC, so their presence should suggest a different diagnosis. Regenerative crypt changes and pyloric metaplasia are not commonly seen in cases of NSAID-related colitis unless patients have taken high doses over an extended period [14,19]. Conversely, patchy neutrophilic cryptitis, apoptotic crypt epithelial cells, increased intraepithelial lymphocytes, and regenerative changes may variably be present, with plasma cells and lymphocytes sparsely distributed in the lamina propria [14,19].

### 5.5. Focal Active Colitis

The inflammatory pattern of FAC is associated with bacterial infections (Campylobacter, Salmonella, Shigella, *E. coli*, and Yersinia), viral infections, or drug-induced injury [14]. Other clinical associations include IBS and IBD, particularly CD [14]. In the absence of clinical colitis, these changes often remain unexplained, and could be related to bowel preparation [14]. On the other hand, diffuse active colitis is somewhat specific to the active phase of ulcerative colitis, although it is also present in some cases of CD with colonic involvement, infectious colitis, and diverticular disease [14].

As mentioned earlier, FAC is characterized by neutrophilic cryptitis that involves a few contiguous crypts within one biopsy fragment, frequently with increased cellularity of the surrounding lamina propria [14]. The background mucosa shows normal crypt architecture in the absence of Paneth cell metaplasia. When apoptotic crypt epithelial cells are numerous, this finding could suggest a drug-related injury [14].

### 5.6. Pseudomembranous Colitis

The main characteristic of PMC is the presence of laminated pseudomembranes composed of fibrin-rich exudates and mucus embedded with neutrophils and necrotic epithelial cells [14]. These exudates appear to originate from the underlying crypts, which are often dilated, filled with sloughed epithelial cells, neutrophils, or mucus, and with the surface epithelium in contact with the pseudomembrane showing signs of inflammation or necrosis [14]. As the disease progresses, typical ischemic changes occur, initially involving the luminal half of the crypt and eventually extending to affect the entire crypt [14]. The mucosa closer to the pseudomembrane may exhibit features of AC, as it seems relatively spared, or display varying degrees of hemorrhage within the lamina propria [14]. However, a pseudomembranous pattern of damage can also be present in noninfectious ischemic colitis. In noninfectious ischemia, the lesions tend to be localized, and on colonoscopic examination, the pseudomembranes may appear larger, resembling polypoid mass-like lesions. Therefore, if clinical information and endoscopic appearance are not considered, mucosal sampling from these lesions may be misinterpreted as suggestive of C. difficile-associated colitis. Furthermore, hyalinization of the lamina propria and regenerative crypts, along with sparse inflammation, could better indicate a diagnosis of ischemia [20].

### 5.7. Hemorragic Colitis

Among the causes of hemorrhagic colitis, the Shiga-producing enterohemorrhagic strain of *E. coli*, O157:H7, is the best-known [14]. This microorganism primarily colonizes the right colon, resulting in high-volume, non-bloody watery diarrhea that may progress to bloody diarrhea. In approximately 10% of patients, it may be associated with uremic syndrome and thrombotic thrombocytopenic purpura [14,21,22,23]. Unfortunately, killing the bacteria with antibiotic therapy triggers the release of Shiga toxin, which leads to the development of hemolytic uremic syndrome [14,24]. Furthermore, Klebsiella oxytoca, a bacterium that was considered a simple commensal for several years, has been found to be another cause of hemorrhagic colitis [14]. K. oxytoca typically causes segmental colitis of the proximal colon, which usually goes into remission shortly after discontinuation of antibiotic therapy [25]. Finally, other causes of hemorrhagic colitis include Shigella infection, noninfectious acute ischemia, and certain medications, such as some antibiotic agents, alpha-interferon, hyperosmolar medication formulations, and other drugs [26].

In this setting, the lamina propria is edematous and hemorrhagic [14]. Moreover, fibrin thrombi within capillaries are usually observed [14]. Crypts show ischemic injury and/or display a regenerative appearance [14]. There is variable neutrophilic inflammation [14]. Pseudomembranes may be present, although they are not the most prominent feature [14].

### 5.8. Ischaemic Colitis

Acute ischemic colitis is associated with several conditions, including vascular occlusive disorders such as thromboemboli, vascular injury during surgery or trauma, vasculitis, and vascular compromise associated with volvulus, intussusception, and serosal adhesions [14]. Furthermore, non-occlusive etiologies include cardiac failure, shock, sepsis, medications, certain drugs, and infections [14]. Colonoscopy is frequently performed to confirm the diagnosis and assess the severity of ischemia [14]. Mucosal friability with bleeding, hemorrhages, edema, erythema, erosions, and linear ulcers can be observed in various combinations [14].

In more severe cases, there is a loss of haustral markings, cyanosis, and gangrene [14,27]. Ischemic injury can occur at any location within the colon with similar frequency, as recently demonstrated [14,28,29]. Ischemic necrosis initially involves the superficial epithelium up to the luminal third of the crypts. In the early stages of ischemia, the basal portions of the crypts are preserved, although total crypt loss develops with progressive injury [14]. Ischemic crypts tend to have a smaller diameter than their non-ischemic counterparts, a feature referred to as microcrypts or withering crypts [14]. As mentioned earlier, pseudomembranes can also be present. Modifications within the lamina propria include hyalinization, edema, hemorrhage, and scant inflammation.

## 6. Conclusions

FAC is an inflammatory pattern that may or may not be clinically relevant. It is most observed in adult patients with bowel preparation artifacts [9,10,29,30,31,32,33], infectious colitis, drug-induced gastrointestinal injuries, and IBS. Less frequently, it may manifest as IBD [34,35], such as CD or partially treated UC. Since IBS [36,37,38] and drug-induced colitis are uncommon in children, a higher number of pediatric patients with this inflammatory pattern in colonic biopsies are likely to have IBD.

There is still much to uncover, and new data are essential for a better understanding of the extent of FAC. Ongoing research is crucial to further elucidate the active and substantial need for additional data to refine our knowledge.

In recent years, the correlation between the microbiota and colon inflammation has been supported by extensive data [39,40,41,42], as observed in other gastrointestinal conditions [43,44,45]. Defining the role of the microbiota in this pathological condition represents a new research direction for understanding the disease’s mechanism. Furthermore, detecting earlier lesions in other locations of the gastrointestinal tract could be helpful to confirm an early diagnosis of IBD in the presence of focal active colitis [46,47,48].

## Figures and Tables

**Table 1 biomedicines-11-02631-t001:** Diagnoses associated with the occurrence of FAC in the available studies regarding clinical implications of FAC in the adult setting.

Author, Year	Study Lenght	No. Patients	Infections	Drugs	Irritable Bowel Syndrome	Incidental	Inflammatory Bowel Disease	Other
Greenson, 1997 [2]	1988–1993	42	55%	45%	14%	26%	0%	5%
Volk, 1997 [4]	1982–1992	31	48%	?	?	29%	13%	10%
Shetty, 2011 [3]	1994–2004	90	19%	24%	33%	8%	16%	-
Sinagra, 2017 [5]	2012–2016	30	3%	30%	27%	17%	23%	-
Hosack, 2022 [6]	2014–2019	43	33%	7%	19%	26%	12%	2%

**Table 2 biomedicines-11-02631-t002:** Diagnoses associated with the occurrence of FAC in the available studies regarding clinical implications of FAC in the pediatric setting.

Author, Year	Study Lenght	No. Patients	Infections	Drugs	Irritable Bowel Syndrome	Incidental	Inflammatory Bowel Disease	Other
Xin, 2003 [7]	1989–2000	31	31%	0%	0%	28%	31%	10%
Osmond, 2018 [8]	2002–2015	68	3%	-	1%	48%	24%	9% (allergic colitis)

## Data Availability

Not applicable.

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
