# Peer review of "Focal Active Colitis: What Are Its Clinical Implications? A Narrative Review"

_biomedicines, 2023, doi:10.3390/biomedicines11102631_

Round 1

Reviewer 1 Report

This is a review on the evidence of the clinical implications of FAC, both in the adult and in the pediatric population (total of 7 studies available from the literature). It is an interesting read. Here are some comments for the authors:

1. Tables 1 and 2 are very informative. Authors should consider writing in a different colum in tables 1 and 2 the location of the studies and their length eg.: 1989-2000.

2. irritable bowel syndrome could be replaced with IBS (see whole text), the same with IBD.

1. Authors should consider removing the 'et al.' from the senetnces like: "According to Greenson et al. and coworkers..." etc.

2.  Minor editing of English language is required

Author Response

Response to reviewers’ comments

Dear Editors, dear Reviewers,

We wish to express our appreciation to the Editors and Reviewers for their insightful comments, which have helped us significantly to improve our manuscript. According to the suggestions, we have thoroughly revised our manuscript and its final version is enclosed. Point-by-point responses to the comments are listed below.

Reviewers’ comments #1

- Tables 1 and 2 are very informative. Authors should consider writing in a different colum in tables 1 and 2 the location of the studies and their length eg.: 1989-2000.

Response: Dear reviewer, many thanks for Your valuable comment ; we corrected tables accordingly

- irritable bowel syndrome could be replaced with IBS (see whole text), the same with IBD.

Response: Dear reviewer, many thanks for Your valuable comment ; we spelled the acronyms accordingly

- Authors should consider removing the 'et al.' from the senetnces like: "According to Greenson et al. and coworkers..." etc.

Response: we corrected these sentences accordingly

- Minor editing of English language is required

Response: manuscript was edited by a native english speaker

Reviewer 2 Report

 Focal Active Colitis (FAC): which are its clinical implications? A narrative review

I have read above manuscript with interest and care.

 Several points are requested;

1.     Figures are not attached.

2.     Some request as follows;

Aim of this review is clinical implication of FAC. One of important significance is correlation of Crohn’s disease (CD). FAC can be stigmata or precursor lesion of CD or not. It is a good point and there are a lot of studies to detect earlier lesions. Then diagnostic process had been constructed with endoscopic and histologic findings. Thus, how to construct diagnostic process should be the one of important issue in this article. We need active recommendations how to construct such process.

Reference

1.     Increased numbers of macrophages in noninflamed gastroduodenal mucosa of patients with Crohn's disease. K Yao A IwashitaT Yao, et al.  Dis Sci1996; 41(11):2260-7.

2.     Incidence, clinical characteristics, long-term course, and comparison of progressive and nonprogressive cases of aphthous-type Crohn's disease: a single-center cohort study. Tsurumi K, Matsui T, Hirai F, et al. Digestion. 2013; 87(4):262-8.

3.     Focally enhanced gastritis: a frequent type of gastritis in patients with Crohn's disease. Oberhuber G, Püspök A, Oesterreicher C, et al. Gastroenterology. 1997;112(3):698-706.

no

Author Response

Response to reviewers’ comments

Dear Editors, dear Reviewers,

We wish to express our appreciation to the Editors and Reviewers for their insightful comments, which have helped us significantly to improve our manuscript. According to the suggestions, we have thoroughly revised our manuscript and its final version is enclosed. Point-by-point responses to the comments are listed below.

Reviewers’ comments #2

- Figures are not attached.

Response: Dear reviewer, we attached in a former version of the manuscript, 2 figures, but since they were original, Editor did not allow to publish them in a review article

- Aim of this review is clinical implication of FAC. One of important significance is correlation of Crohn’s disease (CD). FAC can be stigmata or precursor lesion of CD or not. It is a good point and there are a lot of studies to detect earlier lesions. Then diagnostic process had been constructed with endoscopic and histologic findings. Thus, how to construct diagnostic process should be the one of important issue in this article. We need active recommendations how to construct such process.

- Response: dear reviewer, many thanks for Your valuable comment. We added in the “clinical implication” section, we added the following paragraph, according to Your suggestion: “The FAC pattern is typically not observed in UC; when it is present it raises suspicion for a diagnosis of Crohn’s colitis or infectious colitis and/or acute self-limited colitis. In particular, FAC can be stigmata or precursor lesion of CD or not. As aforementioned, interestingly, Hosack and coworkers concluded that having normal FC levels and endoscopic findings reduces the risk of future development of IBD in adults, while higher FC levels and endoscopic features suggestive of IBD with histological FAC may predict progression to IBD. On the other hand, in the pediatric setting, since IBS and drug-induced colitis are uncommon in children, a higher number of pediatric patients with the inflammatory pattern of FAC in colonic biopsies are likely to have IBD.”

Furthermore, we added the following references

Reference

1. Increased numbers of macrophages in noninflamed gastroduodenal mucosa of patients with Crohn's disease. K Yao , A Iwashita, T Yao, et al. Dis Sci. 1996; 41(11):2260-7.

2. Incidence, clinical characteristics, long-term course, and comparison of progressive and nonprogressive cases of aphthous-type Crohn's disease: a single-center cohort study. Tsurumi K, Matsui T, Hirai F, et al. Digestion. 2013; 87(4):262-8.

3. Focally enhanced gastritis: a frequent type of gastritis in patients with Crohn's disease. Oberhuber G, Püspök A, Oesterreicher C, et al. Gastroenterology. 1997;112(3):698-706.US-GEA, LAMS and SEMS

Finally, we edited the text after a review by a native English speaker

Many thanks again

Sincerely Yours

Emanuele Sinagra

Round 2

Reviewer 2 Report

in page 2 still depicted fihure 1 and 2, which shoukd deleted.

Other part is rewritten and understandable according to revewer's commnts.. Then this manuscript is acceptable.

Betted rewritten.

Author Response

Dear reviewer, we corrected the manuscript accordingly